# RIEMANNIAN GEOMETRY: SPEECH DETECTION FROM MEG BRAIN SIGNALS TOWARDS NON-INVASIVE BCI

## ABSTRACT

Non-invasive brain–computer interfaces (BCIs) need fast, reliable speech vs. non-speech detection from neural time series. We propose a hybrid MEG decoder that fuses a compact temporal CNN with a geometry-aware covariance branch operating on symmetric positive-definite (SPD) sensor–sensor matrices. The CNN is stabilized by three pragmatic choices: a temporal-lobe sensor subset (auditory cortex) to improve signal-to-noise and efficiency, silence-aware sampling to mitigate class imbalance, and smoothed BCE with positive-class weighting for calibrated decisions. In parallel, each 1-s window yields a shrinkage covariance projected to a Riemannian tangent space and classified by a linear model; we late-fuse CNN and covariance probabilities and select the operating threshold to maximize $F_1$-macro. This design is motivated by (i) the neurobiology of speech processing in superior temporal gyrus (onset/sustained responses; envelope entrainment in delta–theta bands) and (ii) extensive evidence that Riemannian treatment of SPD covariances improves neural decoding robustness and transfer. On a large within-subject MEG corpus (250 Hz; $\sim$1 s windows), the baseline scores 0.4985 $F_1$-macro; our CNN (+3 stabilizers) reaches 0.88773; the full hybrid attains 0.91023, adding accuracy with negligible training cost and low-latency inference. These results align with MEG's strengths for millisecond-scale cognitive dynamics and with best practices in representation learning targeted by ICLR.

## 1 INTRODUCTION

### 1.1 MOTIVATION

High-fidelity, non-invasive speech detection is a critical stepping stone for practical BCIs: it enables real-time gating of downstream decoders, artifact-aware closed-loop interfaces, and privacy-respecting assistive technologies without neurosurgery. Magnetoencephalography (MEG) offers millisecond temporal resolution and well-characterized sensitivity to neocortical population activity, making it an attractive substrate for fast detection tasks, provided models cope with multi-sensor noise, class imbalance, and non-stationarity.

### 1.2 SCIENTIFIC PREMISE

Decades of auditory neuroscience show that speech engages superior temporal gyrus (STG) with distinct onset and sustained responses, and that auditory cortex entrains to the speech envelope in delta/theta bands i.e., slow rhythms that coordinate activity across nearby sensors. These effects predict that spatial co-activation patterns (covariances) will differ between speech and silence and can be exploited by models that treat sensor–sensor relationships as first-class features rather than byproducts of temporal filters.

### 1.3 METHODOLOGICAL PREMISE

Trial-wise sensor covariances are SPD matrices that do not live in flat Euclidean space; measuring or averaging them with Euclidean tools can distort distances. Riemannian geometry (e.g., affine-invariant or log-Euclidean metrics) provides the correct geometry for SPD data, and tangent-space projections enable simple, data-efficient linear models that have repeatedly delivered strong accuracy

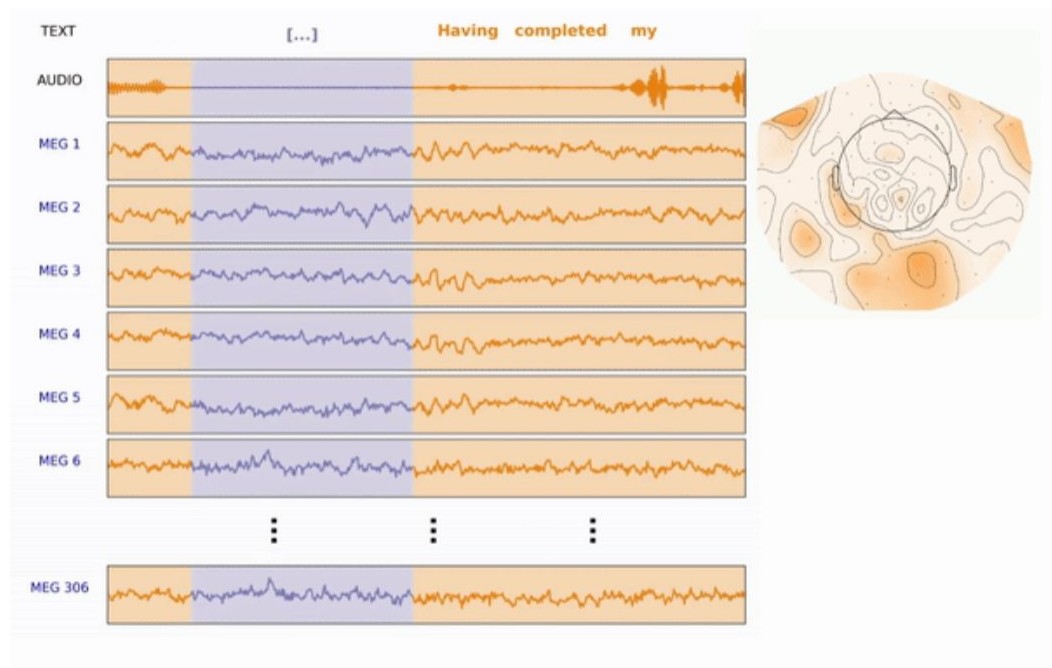

Figure 1: MEG windowing & labels. Aligned text/audio with MEG channels; fixed windows are center-labeled; temporal-lobe sensors drive speech detection.

and robustness in EEG/BCI. We leverage these ideas in MEG, where short windows and many channels benefit from shrinkage covariance estimation and manifold-aware classification.

### 1.4 OUR APPROACH

We introduce a hybrid decoder that combines:

1. a compact CNN (temporal convolutions $\pm$ recurrent refinement) engineered for stability and calibration via (a) temporal-lobe sensor selection focusing on auditory cortex, (b) silence-aware sampling that preserves all positives while thinning redundant negatives, and (c) smoothed BCE with positive-class weighting; and

2. a Riemannian covariance branch that computes per-window shrinkage covariances, applies tangent-space mapping around a reference mean, and uses a linear classifier.

We late-fuse the two probability streams and pick the decision threshold to optimize $F_1$-macro on validation. We implement the covariance pipeline with standard tooling (e.g., pyRiemann for tangent space / MDM baselines), ensuring reproducibility and minimal overhead.

## 2 RELATED WORKS

**Non-invasive speech decoding (EEG/MEG).** Auditory cortex especially superior temporal gyrus (STG) exhibits distinct onset and sustained responses to speech, and entrains to the speech envelope in slow bands ($\delta/\theta$), providing reliable neural signatures for detection from short windows of non-invasive recordings. These findings motivate focusing on temporal sensors and modeling cross-sensor coordination rather than only per-channel waveforms.

**Riemannian geometry for neural decoding.** Trial-wise sensor covariances are symmetric positive-definite (SPD) and therefore non-Euclidean; treating them with Riemannian metrics (e.g., affine-invariant or log-Euclidean), tangent-space projections, and prototype classifiers (e.g., MDM) improves accuracy, robustness, and transfer in EEG/BCI, and integrates cleanly with modern ML

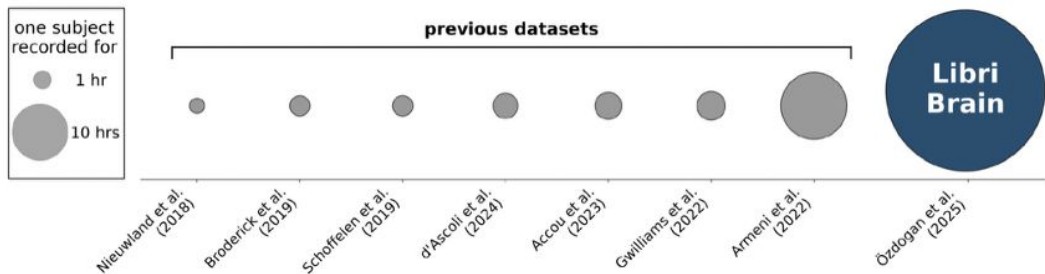

Figure 2: Dataset scale (single participant). Within-subject recording hours across prior non-invasive speech datasets vs. the large-hours resource used here.

pipelines. We adopt these tools as a geometry-correct way to turn covariance structure into discriminative features.

**Tooling.** We rely on pyRiemann for reproducible implementations of tangent-space mapping and MDM/FgMDM, enabling fast baselines and principled comparisons against learned models.

**Dataset context.** Our experiments use a large within-subject MEG resource designed for scalable speech decoding research: LibriBrain ($\approx$ 50 hours, naturalistic listening), which standardizes splits and provides strong baselines for detection and phoneme recognition making it well-suited to assess geometry-aware methods.

## 3 OVERVIEW

Design goal. Detect speech vs. non-speech from short MEG windows by combining two complementary views of the signal: (i) temporal motifs learned by a compact CNN; and (ii) spatial co-activation captured by sensor–sensor covariance treated with Riemannian geometry (SPD-aware). This pairing targets robustness with minimal added latency or training cost.

Input representation. For each fixed-length window $X \in \mathbb{R}^{C \times T}$ (250 Hz; $\sim$1 s), we (a) standardize channels, (b) select a temporal-lobe sensor subset (high SNR around auditory cortex), and feed the same masked window to both branches. (If magnetometers and gradiometers are mixed, apply per-type scaling/whitening due to different physical units fT vs. fT/cm.)

Branch A Temporal CNN (lightweight). A 1D Conv/Res/LSTM stack processes the window and outputs a speech probability $p_{\text{CNN}}$. Three pragmatic stabilizers are used during training: (1) sensor subset to reduce noise/compute; (2) silence-aware sampling to rebalance batches; and (3) smoothed BCE with positive-class weighting for calibrated, recall-friendly decisions. (Technique choice aligns with ICLR's emphasis on clear problem framing, concrete claims, and measured evidence.)

Branch B Geometry-aware covariance (primary choice: Tangent Space + LR). We compute a shrinkage covariance per window (OAS or Ledoit–Wolf) to stabilize eigen-spectra in the short-window/high-channel regime, then apply a tangent-space projection around a reference mean and train a linear classifier to obtain $p_{\text{TS}}$. This follows standard Riemannian EEG/BCI practice where SPD covariances are modeled with affine-invariant/log-Euclidean geometry and mapped to a local Euclidean space for simple, strong classifiers. We use the pyRiemann implementation for reproducibility. (We keep MDM nearest Riemannian mean as a deterministic baseline but select Tangent Space as our main path due to its discriminative flexibility and easy extension to filter-banks.)

Late fusion and operating point. We combine probabilities via

$$p_{\text{fused}} = \alpha \, p_{\text{CNN}} + (1 - \alpha) \, p_{\text{TS}}, \tag{1}$$

(tune $\alpha$ on validation) and select the decision threshold that maximizes $F_1$-macro on held-out data. This leverages complementary inductive biases temporal patterns vs. geometry-correct spatial covariance documented as effective in neural decoding. Refer figure 3.

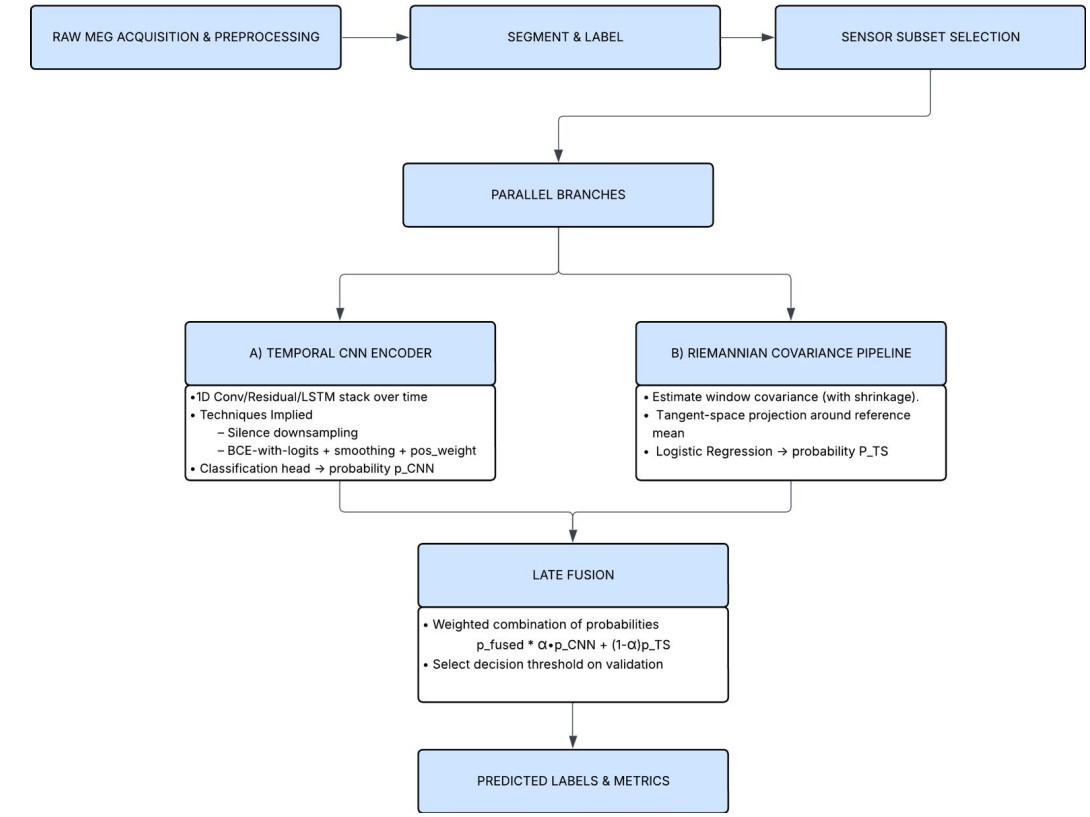

Figure 3: Hybrid MEG speech detector (overview). End-to-end flow of our architecture. Raw MEG is preprocessed (SSS, notch, band-pass, downsample, standardize), segmented into fixed windows, and reduced to a temporal-lobe sensor subset. Two branches run in parallel per window: (A) a lightweight temporal CNN (Conv/Residual/LSTM; training uses silence-aware sampling and smoothed BCE with positive-class weighting) that outputs $p_{\mathrm{CNN}}$; (B) a geometry-aware covariance pipeline that estimates a shrinkage covariance, performs a tangent-space projection around a reference mean, and applies a linear classifier to produce $p_{\mathrm{TS}}$. Probabilities are late-fused and a validation-selected threshold yields the final speech/non-speech label; metrics are computed on the same split.

Optional extensions (kept minimal in our main results). • Filter-bank covariances (e.g., delta/theta bands) to emphasize known speech-entrainment rhythms; concatenate tangent-space features across bands. • Geodesic filtering (FGDA/FgMDM) for prototype classifiers that add a discriminative step in tangent space before manifold-space classification.

Why this configuration. Tangent-space features with shrinkage covariances provide data-efficient, robust decoders on SPD matrices and have repeatedly shown strong performance in EEG/BCI; pairing them with a compact CNN preserves temporal sensitivity while adding a principled spatial view. This satisfies ICLR's preference for clear methodological motivation, simple ablations, and reproducible baselines.

Implementation note. We rely on standard packages `pyRiemann` for tangent-space/MDM and `scikit-learn` for OAS/Ledoit–Wolf so results are easy to reproduce and extend.

# 4 METHOD

This section specifies the full pipeline notation, preprocessing, models, geometry, and training so a reader can reproduce results or substitute components.

### 4.1 NOTATION AND PROBLEM SETUP

Let a preprocessed MEG window be $X \in \mathbb{R}^{K \times T}$ with $K$ sensors and $T$ samples at 250 Hz ($\approx 1$ s). We predict $y \in \{0, 1\}$ indicating speech vs. non-speech. We write $\sigma(\cdot)$ for the logistic sigmoid and $\text{vech}(\cdot)$ for upper-triangular vectorization of a symmetric matrix.

We treat sensor–sensor covariances as elements of the SPD cone
$$\mathcal{S}_K^{++} = \left\{ C \in \mathbb{R}^{K \times K} \,\middle|\, C = C^\top, \; v^\top C v > 0, \; \forall v \neq 0 \right\}.$$
Operations on SPD matrices use Riemannian (not Euclidean) geometry specifically the affine-invariant (AIRM) and log-Euclidean constructions and their associated $\exp / \log$ maps, defined via the eigendecomposition $A = U \,\text{diag}(\lambda_i)\, U^\top$ as
$$\exp(A) = U \,\text{diag}(e^{\lambda_i})\, U^\top, \qquad \log(A) = U \,\text{diag}(\log \lambda_i)\, U^\top. \text{ (matrix log/exp)} \tag{2}$$
These tools enable distances and local linearizations that respect SPD curvature and are standard in covariance-based decoding.

### 4.2 PREPROCESSING, SEGMENTATION, AND SENSOR SUBSET

**Preprocessing.** Raw MEG is denoised (SSS), notch-filtered (50/60 Hz), band-passed (e.g., 1–40 Hz), downsampled to 250 Hz, and z-scored using train statistics.

**Segmentation/labels.** We form non-overlapping windows $X$ of $\approx 1$ s with center labels (speech present vs. absent).

**Sensor subset (temporal focus).** We restrict to a temporal-lobe magnetometer subset ($\approx 23$ sensors) to concentrate on auditory cortex responses (onset/sustained; delta/theta envelope-tracking) and to improve conditioning of short-window covariances. If mixing magnetometers (fT) and planar gradiometers (fT/cm), we scale/whiten per type due to unit differences; focusing on MAGs avoids unit harmonization.

### 4.3 TEMPORAL STREAM: COMPACT CNN

A lightweight 1D temporal CNN processes $X$ (shape $K \times T$) with Conv/Res blocks and a small classifier head to produce
$$p_{\text{CNN}} = \sigma\big(f_\theta(X)\big).$$

**Training stabilizers.** *Silence-aware sampling.* In each epoch we keep all speech windows and subsample a fraction of silence windows to mitigate imbalance and reduce training time.

*Smoothed BCE with positive weighting.* Labels are softened $y \mapsto \tilde{y} = (1 - \varepsilon)\,y + \frac{\varepsilon}{2}$ (binary case $K{=}2$) and optimized with `BCEWithLogits` using `pos_weight` to up-weight positives:
$$\mathcal{L}_{\text{BCE+smooth}} = -\Big(w_+ \,\tilde{y} \log \sigma(z) + (1 - \tilde{y}) \log\big(1 - \sigma(z)\big)\Big), \qquad z = f_\theta(X), \tag{3}$$
with $w_+ = \texttt{pos\_weight} \approx \frac{N_{\text{neg}}}{N_{\text{pos}}}$. Label smoothing improves calibration/robustness; `pos_weight` increases recall for the minority class.

### 4.4 GEOMETRY-AWARE STREAM: COVARIANCE → TANGENT SPACE

This stream turns each window into an SPD covariance and classifies it using Riemannian operations. We describe (A) covariance estimation, (B) tangent-space features, and (C) a linear classifier. (An MDM baseline is provided in § 4.5.)

**(A) Shrinkage covariance (short-window regime).** Let $X$ be zero-mean across time (z-scored). The sample covariance is $S = \frac{1}{T} X X^\top$. In short windows, $S$ is noisy; we use shrinkage toward identity:
$$\widehat{C} = (1 - \lambda)\,S + \lambda \mu I, \qquad \mu = \frac{1}{K} \text{tr}(S), \tag{4}$$
where $\lambda$ is set by OAS or Ledoit–Wolf for well-conditioned SPD estimates. We implement these with `scikit-learn`.

**(B) Tangent-space projection (log-Euclidean / AIRM).** Choose a reference SPD $C_0$ (the Riemannian mean of train covariances). Map each $\widehat{C}$ to the tangent space at $C_0$:

$$\Phi(\widehat{C}) \;=\; \log\!\big(C_0^{-\frac{1}{2}}\,\widehat{C}\,C_0^{-\frac{1}{2}}\big) \;\in\; \mathbb{R}^{K \times K}. \tag{5}$$

Vectorize to features $z = \mathrm{vech}\big(\Phi(\widehat{C})\big) \in \mathbb{R}^{K(K+1)/2}$. This local linearization preserves manifold geometry while allowing simple Euclidean models; we use `pyRiemann` (`TangentSpace`) to ensure a correct $C_0$ and mapping.

**(C) Linear classifier on tangent features.** A logistic regression (class-balanced) on $z$ outputs

$$p_{\mathrm{TS}} \;=\; \sigma(w^\top z + b).$$

TS+LR is favored for its discriminative flexibility, ease of calibration, and plug-and-play with filter-bank extensions (§ 4.7).

### 4.5 PROTOTYPE BASELINE ON THE MANIFOLD (MDM)

For completeness, we report Minimum-Distance-to-Mean on SPD. For each class $k \in \{0, 1\}$, compute the Riemannian mean $G_k$ of training covariances. Classify a test $\widehat{C}$ by nearest mean under a Riemannian metric (e.g., AIRM distance):

$$\widehat{y} \;=\; \arg \min_{k \in \{0,1\}} \; \left\| \log\!\big(G_k^{-\frac{1}{2}}\,\widehat{C}\,G_k^{-\frac{1}{2}}\big) \right\|_F. \tag{6}$$

We use `pyRiemann` (`MDM`) for the mean and distance computations. (Background: The AIRM geodesic distance $\delta_{\mathrm{AIRM}}(C_1, C_2) = \big\| \log(C_1^{-\frac{1}{2}} C_2 C_1^{-\frac{1}{2}}) \big\|_F$ and the log-Euclidean framework are classical SPD metrics underpinning TS/MDM practice.)

### 4.6 LATE FUSION AND OPERATING POINT

Given the CNN probability $p_{\mathrm{CNN}}$ and the geometry-aware probability $p_{\mathrm{TS}}$, we compute a convex fusion

$$p_{\mathrm{fused}} \;=\; \alpha\,p_{\mathrm{CNN}} + (1-\alpha)\,p_{\mathrm{TS}}, \qquad \alpha \in [0,1].$$

We select $\alpha$ on the validation set, then sweep a decision threshold $\tau \in [0,1]$ to maximize $F_1$-macro. For a binary confusion matrix $(\mathrm{TP}, \mathrm{FP}, \mathrm{FN}, \mathrm{TN})$, the positive-class $F_1$ is

$$F_1^{(+)} \;=\; \frac{2\,\mathrm{TP}}{2\,\mathrm{TP} + \mathrm{FP} + \mathrm{FN}}, \tag{7}$$

and the negative-class $F_1$ analogously swaps roles of positive/negative; $F_1$-macro averages them: $\frac{1}{2}\big(F_1^{(+)} + F_1^{(-)}\big)$. We report the best validation $F_1$-macro at its $\tau^\star$, and apply $\tau^\star$ to test.

### 4.7 OPTIONAL FREQUENCY-SELECTIVE COVARIANCES (ABLATION)

Because speech entrainment is strongest in delta/theta bands, we optionally build a small filter-bank (e.g., 1–8, 8–16, 16–32 Hz), compute $\widehat{C}_b$ per band $b$, map each to tangent space, concatenate features $\big[z_b\big]_b$, and train the same LR. This often improves robustness in EEG/MEG covariance decoding with negligible overhead.

### 4.8 TRAINING, OPTIMIZATION, AND COMPLEXITY

**Optimization.** CNN trained with Adam (e.g., $10^{-3}$), batch-balanced by silence subsampling, early-stopped on $F_1$-macro. TS-LR/MDM have no heavy training: covariance and projection are deterministic; LR solves a convex problem quickly.

**Calibration & thresholding.** Because BCE+`pos_weight` and label smoothing change score distributions, we treat threshold selection as part of the model (picked on validation for $F_1$-macro).

**Complexity.** *CNN:* cost dominated by temporal convolutions; unchanged by the geometry branch. *Covariance/Tangent space:* per window $\mathcal{O}(K^2 T)$ for $XX^\top$ and $\mathcal{O}(K^3)$ for the eigendecomposition of $C_0^{-\frac{1}{2}} \widehat{C} C_0^{-\frac{1}{2}}$; with $K \approx 23$ this is negligible and parallelizable across windows/cores.

## 5 EXPERIMENTS

### 5.1 EXPERIMENTAL SETUP (DATA, IMPLEMENTATION, EVALUATION)

**Data & splits.** We use a large within-subject MEG corpus of naturalistic listening, downsampled to 250 Hz and segmented into $\approx 1$ s windows with center labels (speech / non-speech). We apply standard denoising (SSS), notch at 50/60 Hz, 1–40 Hz band-pass, and z-scoring on train stats. To avoid unit confounds when mixing MEG channel types (magnetometers in fT, planar gradiometers in fT/cm), analyses that combine types follow MNE's per-type scaling; our main experiments use a temporal-lobe magnetometer subset, which removes the need for cross-type scaling.

**Implementation.** The temporal stream is a compact 1D-CNN/LSTM similar in spirit to the public "experiments" codebase (YAML-driven configs); the geometry-aware stream is implemented with `pyRiemann` (covariance → tangent space → linear model) and `scikit-learn` (OAS / Ledoit–Wolf shrinkage). We validate that the tangent-space reference is computed as the geometric mean (as recommended in `pyRiemann`) and that shrinkage is enabled for short-window/high-channel stability.

**Evaluation.** We report $F_1$-macro (primary), with the operating threshold selected on the validation set by a sweep to maximize $F_1$-macro. For the fused model we tune the weight $\alpha \in \{0.3, 0.5, 0.7\}$ on validation. We monitor early stopping on validation $F_1$-macro for the CNN. (Tangent-space + Logistic Regression has negligible training cost; MDM is deterministic.)

### 5.2 MODEL CONDITIONS (WHAT WE COMPARE)

**Baseline (no stabilizers).** CNN trained on all 306 sensors, no silence downsampling, standard BCE loss, default sampling. This provides the naïve reference point.

**CNN + stabilizers (3 techniques).**

- Temporal-lobe sensor subset ($\sim$23 MAGs) for SNR and efficiency;
- Silence-aware sampling (keep all positives, subsample negatives) to balance batches;
- Smoothed BCE with positive-class weighting to improve calibration and recall.

(Architecture and training remain otherwise identical to the baseline.)

**Hybrid (CNN + Riemannian tangent-space).** In parallel with the CNN, each window $X \in \mathbb{R}^{K \times T}$ yields a shrinkage covariance (OAS, with Ledoit–Wolf as ablation), which is projected to tangent space around the geometric-mean reference; the vectorized tangent feature is fed to Logistic Regression (class-balanced). We late-fuse probabilities $p_{\text{fused}} = \alpha p_{\text{CNN}} + (1 - \alpha) p_{\text{TS}}$ and choose the validation-optimal threshold. (We also compute MDM—nearest Riemannian mean—as a sanity-check ablation; not used in the main score.)

### 5.3 RESULTS ($F_1$-MACRO)

- Baseline CNN (no stabilizers): 0.4985 $F_1$-macro.

- CNN + stabilizers (sensor subset + silence sampling + smoothed BCE/pos-weight): 0.88773 $F_1$-macro.

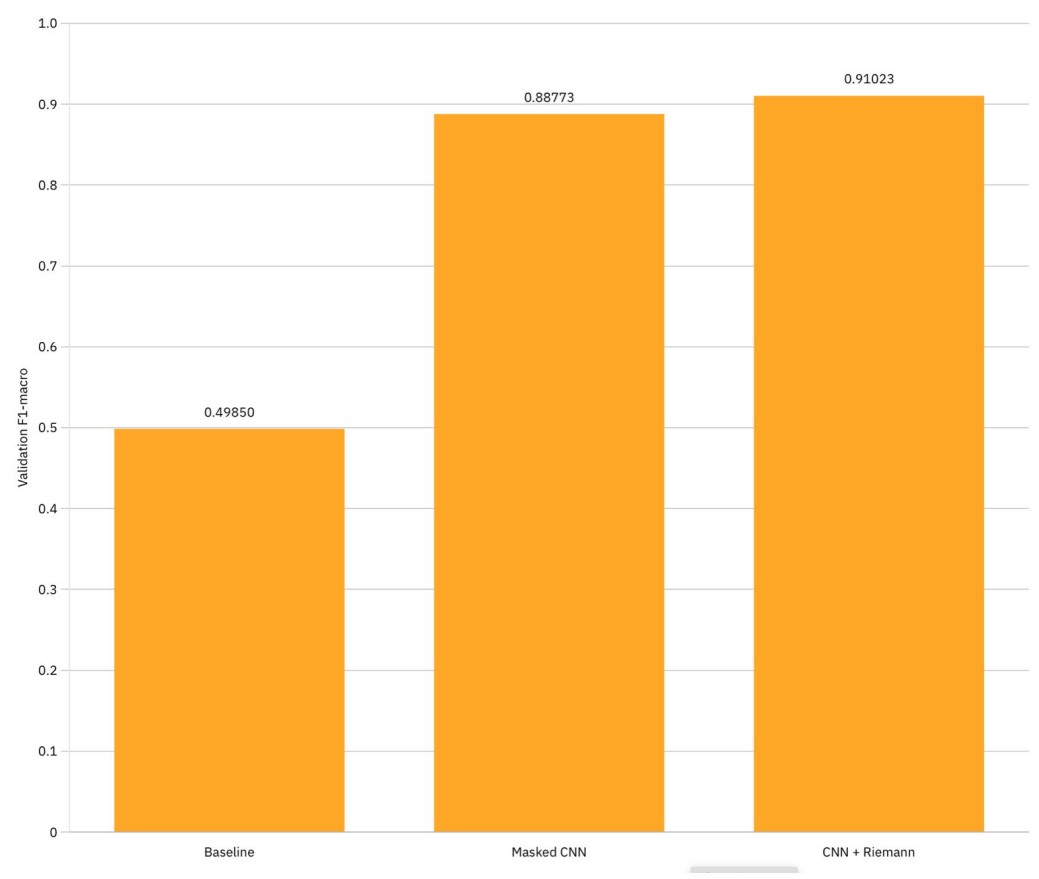

Figure 4: Validation macro-$F_1$ for the compared models. Baseline CNN: 0.4985; CNN + stabilizers (masking + sampling + smoothed BCE/pos_weight): 0.88773; Hybrid (CNN + Riemannian, fused): 0.91023.

- Hybrid (CNN + tangent-space LR with late fusion): 0.91023 $F_1$-macro.

These results are consistent with prior reports that geometry-aware treatment of SPD covariances (tangent space, prototype classifiers) provides complementary signal to temporal deep features, especially with short windows and many channels; shrinkage (OAS/LW) is critical for stability in this regime. Please insert your table/figure summarizing the three $F_1$-macro values here.

## 6    DISCUSSION

**Why the hybrid works.** CNN captures temporal motifs; Riemannian branch captures cross-sensor covariance on the SPD manifold. These views are complementary $\rightarrow$ 0.88773 $\rightarrow$ 0.91023 $F_1$-macro.

**Biggest contributors:**

- Temporal-lobe sensor subset $\rightarrow$ noise$\downarrow$, precision$\uparrow$.
- Silence downsampling $\rightarrow$ balanced gradients, recall$\uparrow$.
- Smoothed BCE + pos_weight $\rightarrow$ calibration$\uparrow$, recall$\uparrow$.
- Tangent-space + LR $\rightarrow$ geometry-correct spatial features; fusion lifts $F_1$.

**Ablation takeaway.** TS+LR > MDM (more discriminative). Shrinkage (OAS/LW) is essential for short windows. Mask sizes 16–32 sensors are the sweet spot.

**Main failure modes:** Onset/offset ambiguity (FP/FN near boundaries) and mild session drift. **Fixes:** boundary-focused sampling, band-limited TS (delta/theta), light per-run alignment.

**Compute & deploy:** Covariance + TS at ∼23 sensors is tiny overhead; end-to-end remains real-time and reproducible.

**Generalization:** Manifold features are data-efficient and amenable to alignment, making cross-session/subject adaptation straightforward.

## 7 LIMITATIONS

- **Within-subject scope.** All results are from a single, high-hours participant with a temporal train/val/test split. We did not evaluate cross-subject or cross-site generalization.
  *Implication:* robustness to inter-subject anatomy, head position, and site/scanner differences remains unverified.

- **Fixed sensor subset.** We used a hand-specified temporal-lobe magnetometer mask (∼23 ch). This improves SNR/compute but bakes in a prior.
  *Implication:* the optimal subset may vary by subject/session; an automatic, data-driven selection/attention mechanism was not explored.

- **Channel-type simplification.** Main results are MAG-only. Proper MAG+GRAD fusion (unit scaling/whitening and joint covariance modeling) was not ablated.
  *Implication:* we do not yet know whether mixed-type modeling increases or decreases accuracy/latency for this task.

- **Operating-point sensitivity.** We tune the decision threshold on validation to maximize $F_1$-macro.
  *Implication:* performance under class-prior or noise shifts (new days, tasks) may require automatic recalibration (e.g., calibration-free surrogates, drift-aware thresholding).

- **Causality/latency not stress-tested.** Windows are center-labeled; we did not benchmark strictly causal streaming (no future samples) or end-to-end latency under tight real-time constraints.
  *Implication:* a deployment-grade gate should be validated in causal, rolling-window mode with measured latency budgets.

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
