# OpenReview forum: "Riemannian Geometry: Speech Detection from MEG Brain Signals Towards Non-Invasive BCI"
_ICLR.cc/2026/Conference — ICLR 2026 Conference Withdrawn Submission_

### Official Review · Reviewer_vWTT · 2025-10-25

**Soundness:** 1
**Presentation:** 1
**Contribution:** 1
**Rating:** 2
**Confidence:** 5

**Summary:**

The paper presents a pipeline for speech detection based on within-subject MEG recordings from a publicly available dataset.

**Strengths:**

The proposed method achieves a slightly better result than the baseline reported in the paper that introduced the dataset (0.91023 vs. 0.8989, Table 3 in [1]).



[1] Özdogan, Miran, et al. "LibriBrain: Over 50 Hours of Within-Subject MEG to Improve Speech Decoding Methods at Scale." arXiv preprint arXiv:2506.02098 (2025).

**Weaknesses:**

1. Figure attribution:

Figure 2 appears to be a direct screenshot or duplicate of Figure 1 from [1]. If this material is reused, the source must be clearly cited in the caption.

2. Comparison with existing results:

The dataset used in this work was part of a completed competition [2], which ended on August 1, approximately two months before the ICLR 2025 submission deadline. Multiple public leaderboard entries [3] already achieved higher performance than the results reported in this paper. However, the authors neither compared their method against these existing results nor provided an explanation for ignoring such comparisons. Therefore, it raises concerns regarding the fairness and completeness of this work.


3. Overall quality and completeness:

The submission appears substantially incomplete, such as pre-processing, model architecture, design rationale, and training details, are missing or only superficially described. As written, the work is not reproducible, and the novelty relative to existing baselines remains unclear.



Refs:

[1] Özdogan, Miran, et al. "LibriBrain: Over 50 Hours of Within-Subject MEG to Improve Speech Decoding Methods at Scale." arXiv preprint arXiv:2506.02098 (2025).

[2] https://neural-processing-lab.github.io/2025-libribrain-competition/participate/

[3] https://neural-processing-lab.github.io/2025-libribrain-competition/leaderboard/speech_detection_standard/

**Questions:**

Please see the comments in the section above.

---

### Official Review · Reviewer_i4zw · 2025-10-30

**Soundness:** 1
**Presentation:** 1
**Contribution:** 1
**Rating:** 0
**Confidence:** 5

**Summary:**

The authors propose a method to detect speech from magnetoencephalography brain waves by fusing a CNN decoder’s probabilities with a classical machine learning model operating on a Riemannian manifold. The suggested advantage is that the latter respects the symmetric positive-definite property of sensor covariance matrices and may provide helpful complementary signal for speech detection.

**Strengths:**

- Late fusion to merge logits and have a geometry-aware deep network architecture is interesting and potentially novel.

**Weaknesses:**

- The title is overclaiming by having “Riemannian Geometry” as the opening clause. Riemannian geometry has been used in various places in neural decoding (see [A] and [B]).
- Similarly, authors have not discussed much of the prior work in neural decoding that uses Riemannian geometry (e.g., particularly [A] and [B]).
- The paper is incomplete, not having cited any prior work (though the references have been filled in)
- Figure 3 can be seriously improved for clarity
- Improvements are marginal and no error bars are provided to determine statistical significance of the result
- The writing is poor, and many parts clearly AI generated and not properly refined and double-checked, e.g. the abstract ends with "these results align with MEG’s strengths for millisecond-scale cognitive dynamics and with best practices in representation learning targeted by ICLR."

Unfortunately, the paper is incomplete and I recommend rejection. I encourage the authors to continue their work by addressing the weaknesses and properly complete the paper as the method is interesting and may have potential to benefit speech decoding, though the evidence currently provided is not convincing.

[A] Huang, Z. and Van Gool, L., 2017, February. A riemannian network for spd matrix learning. In Proceedings of the AAAI conference on artificial intelligence (Vol. 31, No. 1).

[B] Wilson, D., Schirrmeister, R.T., Gemein, L.A. and Ball, T., 2025. Deep riemannian networks for end-to-end eeg decoding. Imaging Neuroscience, 3.

**Questions:**

Why late fuse when the neural network could operate directly on the Riemannian manifold (see [B])?

---

### Official Review · Reviewer_p8bK · 2025-11-06

**Soundness:** 1
**Presentation:** 1
**Contribution:** 1
**Rating:** 0
**Confidence:** 4

**Summary:**

The paper proposes a hybrid MEG-based speech vs non-speech detector combining a lightweight temporal CNN trained with several stabilization heuristics and a geometry-aware covariance branch operating in Riemannian tangent space on SPD matrices.

**Strengths:**

The motivation for non-invasive speech BCIs and the connection to Riemannian geometry are conceptually sound.

**Weaknesses:**

1. Novelty/contribution: The core technical ingredients i.e., tangent-space mapping, shrinkage covariance estimation (Ledoit–Wolf / OAS), logistic regression on SPD features, and late fusion with a CNN are all well-established in EEG/BCI literature. The paper presents an engineering combination of already established ideas rather than a new algorithmic idea, theoretical insight, or learning principle. Using existing Riemannian operations as a plug-in module does not advance our understanding of learning on manifolds or contribute novel methodology to the ML community. The “three stabilizers” (sensor selection, silence sampling, smoothed BCE) are conventional training tricks and cannot be considered contribution.

2. Experimentation/evaluation: Only one subject is evaluated; no cross-subject or cross-session experiments are performed, making generalization claims in Sec 6 unsupported. No variance estimates or statistical significance tests are reported. The experimentation is not rigorous and the claimed +0.02 F1 improvement from fusion may fall within noise. There is no comparison to standard self-supervised or foundation-model baselines (e.g., contrastive, transformer-based MEG encoders). Key hyperparameters (fusion α, smoothing ε, shrinkage λ) are not systematically analyzed. The improvements could stem from better preprocessing rather than the proposed geometry branch.

**Questions:**

Could the authors show cross-subject / cross-day generalization to substantiate claims of robustness in the Discussions section? Also, I think the work will benefit from comparison against SOTA baselines (transformer or graph-based MEG models, self-supervised encoders).

---

### Official Review · Reviewer_5AVv · 2025-11-06

**Soundness:** 2
**Presentation:** 1
**Contribution:** 1
**Rating:** 0
**Confidence:** 4

**Summary:**

This paper proposes a late-fusion MEG decoder for real-time speech vs. non-speech detection for non-invasive brain-computer interfaces (BCIs). The proposed approach leverages established techniques for neural time series data  (e.g. channel selection and use of Riemannian geometry for neural decoding). The authors fit and evaluate their approach on a large single-subject MEG dataset.

**Strengths:**

The choice of methods is well motivated and reasonable for the studied task.

**Weaknesses:**

**Significance**
- The authors study a simple task (speech vs. no-speech, within-subject models) compared to attempting decoding speech from MEG (e.g., Défossez et al., Nat Mach Intell 2023).
- The proposed method is only tested on a single dataset that contains recordings of a single subject.

**Quality**
- Indications of extensive LLM use: No explicit references to prior work in the main text, excessive bullet points, and ICLR-specific promotional sentences like: “(Technique choice aligns with ICLR’s emphasis on clear problem framing, concrete claims, and measured evidence.)” and “This satisfies ICLR’s preference for clear methodological motivation, simple ablations, and reproducible baselines.”
- Imprecise math notation: e.g., equation (2) defines matrix log/exp without defining $\lambda_i$.

**Originality**
- Standard methods used: Loss weighting, simple convolutional networks, feature selection (MEG channels), Riemannian geometry-based classifiers for MEG, and late fusion via convex combination.

**Questions:**

None

---

### Note · Authors · 2026-01-24

I have read and agree with the venue's withdrawal policy on behalf of myself and my co-authors.